# The Effect of the Corrosion Medium on Silane Coatings Deposited on Titanium Grade 2 and Titanium Alloy Ti13Nb13Zr

**DOI:** 10.3390/ma14216350

**Published:** 2021-10-24

**Authors:** Oliwia Kierat, Agata Dudek, Lidia Adamczyk

**Affiliations:** Department of Material Engineering, Faculty of Production Engineering and Materials Technology, Czestochowa University of Technology, Aleja Armii Krajowej 19, 42-200 Czestochowa, Poland; oliwia.kierat@pcz.pl (O.K.); lidia.adamczyk@pcz.pl (L.A.)

**Keywords:** biomaterials, surface modification, silane coatings, corrosion resistance, simulated body fluids

## Abstract

The present paper focuses on the fabrication of coatings based on vinyltrimethoxysilane and the influence of various corrosion media on the coatings produced. Coatings were deposited on two substrate materials, namely, titanium Grade 2 and titanium alloy Ti13Nb13Zr, by immersion in a solution containing vinyltrimethoxysilane, anhydrous ethyl alcohol, acetic acid and distilled water. The obtained coatings were characterized in terms of surface morphology, adhesion to the substrate and corrosion resistance. As corrosion solutions, four different simulated physiological fluids, which differed in the contents of individual ions, and a 1 mol dm^−3^ NaBr solution were used. The chloride ions contained in the simulated physiological fluids did not lead to pitting corrosion of titanium Grade 2 and titanium alloy Ti13Nb13Zr. This investigation shows that titanium undergoes pitting corrosion in a bromide ion medium. It is demonstrated that the investigated coatings slow down corrosion processes in all corrosion media examined.

## 1. Introduction

Titanium is a metallic material which is used in many branches of industry [1]. Titanium alloys are used mainly in aviation, motorization and biomedical engineering [2]. The use of titanium alloys, for instance, in aviation is associated with their valuable properties that make them stand out against other alloys, e.g., aluminum alloys. An extremely valuable feature of titanium alloys from the point of view of the requirements imposed on materials to be used in the aircraft industry is the combination of corrosion resistance, strength, weight and high-temperature stability [3]. Owing to their high biocompatibility, low specific gravity, low elasticity module and good corrosion resistance, titanium and its alloys are used in implantology to substitute for hard tissues [4,5,6,7,8,9]. Until recently, the titanium alloy most commonly used for medical purposes has been a titanium alloy with the addition of aluminum and vanadium—Ti6Al4V [10]. However, studies carried out in recent years have shown that both aluminum and vanadium exert a negative influence on the host’s body [11,12]. In medical applications, titanium alloys containing chiefly elements that are nontoxic and with no allergenic effect, such as niobium, zirconium, tantalum, molybdenum or tin, are being used more and more often [6,10]. Among the main problems involved with the use of metallic materials in implantology, the insufficient corrosion resistance of metals and their alloys is primarily indicated [13,14,15]. The common methods employed in corrosion protection include silanization—a modification of the surface with a silane-based solution [16,17]. Silanes are most commonly used as either coupling agents or crosslinkers [18]. The bonding between the organic silane agent and the inorganic substrate involves the following steps: (i) hydrolysis of the oxide groups of the metallic substrate and the silane coating to form metal hydroxide and silanol; (ii) formation of a hydrogen bond between the metal and silane hydroxyl groups; (iii) condensation of the bonded hydroxyl groups on the substrate and coating surfaces—a Si-O-Me covalent bond is created and the water molecule is released; (iv) condensation of the silane hydroxyl groups—a Si-O-Si siloxane bond is formed and the water molecule is released [13,19]. Among the methods used for depositing coatings on metallic surfaces, electrochemical, chemical and sol–gel methods are distinguished [20]. Notably, the sol–gel method has received special recognition, which is distinguished by a number of advantages [21]. The process of producing materials using the sol–gel method is repeatable, and the process temperature is close to room temperature. A feature that makes the sol–gel method stand out from other methods is also the low cost of its application—it requires no costly apparatus to be used. Among other advantages, one can highlight the ability to produce coatings composed of many components, whose chemical composition is precisely defined, as well as the possibility of producing hybrid materials. Thanks to the use of the sol–gel method, new materials based on silicon can be fabricated [22,23,24]. The sol–gel process consists in the creation of an oxide network as a result of the processing reaction of the condensation of precursors in a liquid medium [20,25]. In the sol–gel process, the hydrolysis and coagulation of precursors take place, resulting in the formation of a sol. Then, the prepared sol is applied to the material by immersing it in the solution. The application of the sol on the material surface is followed by the evaporation of the solvent used for producing the sol, or gelation [21]. After being gelated, the coating can also be cured in a stove.

The purpose of the present study was to produce vinyltrimethoxysilane-based coatings on the substrates of titanium Grade 2 and titanium alloy Ti13Nb13Zr, with the aim of improving their anticorrosive properties for implantology applications. Creating a totally ideal medium rendering the environment occurring in the human body is impossible; however, for the purposes of electrochemical studies aimed at improving the functional properties of implantable biomaterials, simulated physiological solutions are used, which, to the highest possible degree, reflect the human body’s environments [10]. From the chemical point of view, body fluids are electrolytes that differ in their concentrations of potassium, sodium and calcium cations, and chloride and hydrogencarbonate anions. 

## 2. Materials and Methods

All reagents used were analytically pure. For the production of the coating, vinyltrimethoxysilane (VTMS) supplied by Sigma Aldrich (St. Louis, MO, USA) anhydrous ethanol (EtOH) from Chempur (Piekary Slaskie, Poland) acetic acid (AcOH) from Chempur and distilled water were used. The volumetric VTMS:EtOH:AcOH:H_2_O ratio of the obtained coating was 0.6:0.2:0.06:0.14. The total volume of the solution was 10 cm^3^, while the concentration of VTMS was 3.92 mol dm^−3^. The prepared solution was agitated on a magnetic stirrer for 48 hours at a rotational speed of 1000–1500 rpm at room temperature [26].

The solution was applied onto the titanium Grade 2 and titanium alloy Ti13Nb13Zr substrates. The chemical composition of titanium Grade 2 is as follows: Fe—maximum 0.3; O—maximum 0.25; C—maximum 0.08; N—maximum 0.03; H—maximum 0.015; Ti—balanced. Titanium alloy Ti13Nb13Zr has the following chemical composition: Fe—maximum 0.25; O—maximum 0.15; C—maximum 0.08; N—maximum 0.05; H—maximum 0.012; Nb—12.5–14.0; Zr—12.5–14.0; Ti—balanced. For electrochemical tests, samples in the form of 5 mm-diameter cylinders were prepared, which were set in polymethyl methacrylate frames using epoxy resin. Prior to the coating application, each sample was mechanically wet polished using abrasive papers with a grit size of up to 2000. After polishing, all samples were rinsed with distilled water and degreased with acetone. The prepared solution was applied to the prepared samples using the dip coating method. The sample immersion time was 20 min, after which the samples were taken out of the solution, with the excess solution being removed with filter paper. Then, the samples were placed in a desiccator to be thoroughly dried.

Coated samples were analyzed in a 1 mol dm^−3^ NaBr solution that causes pitting corrosion of titanium and its alloys, and in simulated physiological fluids: (i) Ringer’s fluid, (ii) Hank’s fluid, (iii) simulated body fluid and (iv) artificial saliva. The chemical composition of the physiological fluids is shown in Table 1.

Microstructural examination was carried out with a KEYENCE VHX 7000 digital microscope (Keyence, Mechelen, Belgium) and an Olympus GX41 optical microscope (Olympus, Tokyo, Japan). Profilometric examination was conducted with a SENSOFAR profilometer (Sensofar, Barcelona, Spain). The topography of the coatings and their composition were analyzed using a JEOL JSM-6610 LV scanning electron microscope with an EDS X-ray microanalyzer (Jeol, Tokyo, Japan). The characteristics of the coatings were examined with the use of an IRAffinity—1S FTIR SHIMADZU (Kyoto, Japan) spectrophotometer. The adhesion of the coatings to the substrate was tested by the pull-off method, using Scotch^TM^ adhesive tape (Scotch^TM^ Brand, St. Paul, MN, USA). The test involved a sequence of sticking the tape on and then pulling it off the test sample 5 times.

The corrosion behaviors of the biomaterials are shown with potentiodynamic and chronoamperometric curves. Measurements were taken using a CH Instruments 660 measuring station (CH Instruments, Austin, TX, USA) comprising three electrodes: (i) working electrode—the selected titanium substrate; (ii) auxiliary electrode—a platinum electrode; and (iii) reference electrode—a calomel electrode. A potential range from −1.5 to +3.0 V was used for each sample. Potentials were measured with respect to the saturated calomel electrode (SCE).

## 3. Results and Discussion

### 3.1. Characterization of the VTMS Coating

After producing the coatings, a morphology analysis was performed both on the titanium Grade 2 substrate (A) and on the titanium alloy Ti13Nb13Zr substrate (B), as shown in Figure 1. The structure of the metal substrate visible in the photographs is indicative of the transparency of the coating produced. An important asset of the coating is its homogeneity, as well as the absence of cracks and discontinuities on its surface. Moreover, the coating surface is characterized by high gloss.

Figure 2 shows the surface of a coating deposited on titanium Grade 2 (A) and titanium alloy Ti13Nb13Zr (B). The photographs confirm that the applied coating uniformly covers the entire surface of the substrate. Furthermore, no significant differences in coating thickness were noticed on the coated surface. The surface of the VTMS coating deposited on both materials was smooth, uniform and compact.

### 3.2. Thickness and Surface Roughness of the VTMS Coating

Figure 3 shows the cross-section of the produced coating deposited on titanium Grade 2 (A) and titanium alloy Ti13Nb13Zr (B). The presence of the coating was confirmed on the entire sample surface. The thickness of the coating, measured in different locations on the surface, ranged from 10 to 14 μm.

Figure 4 illustrates the profile of the obtained VTMS coating deposited on titanium alloy Ti13Nb13Zr (A) and the uncoated substrate (B). The recorded profile shows that the coating uniformly covers the surface of titanium alloy Ti13Nb13Zr. Both the photographs and the profile confirm that the coating is free of any cracking and its surface roughness is negligible (0.26 μm), which is a huge asset from the point of view of the corrosion resistance of biomaterials. The coating thickness, as measured on the profile base, is about 10 μm. 

In addition, the coat thicknesses were measured using a DT-20 AN 120 157 m (ANTICORR, Gdansk, Poland). Taking into consideration the two substrate materials, recorded thickness values ranged from 9.5 to 16.7 μm, which was also confirmed by examination conducted with a scanning electron microscope and a profilometer.

### 3.3. Chemical Composition of the VTMS Coating

The chemical composition of the produced coating was analyzed with a scanning electron microscope equipped with an EDS-type X-ray microanalyzer. The chemical analysis of the coating revealed the presence of silicon (Si) in the amount of 27.9 wt%. For comparison purposes, the chemical analysis also took into account carbon and oxygen, the contents of which were 41.3 and 30.8 wt%, respectively. 

### 3.4. Characterization of the Structure of the VTMS Coating

The structures of the coatings deposited on titanium Grade 2 and titanium alloy Ti13Nb13Zr were analyzed using Fourier transform infrared spectroscopy (FTIR). Figure 5 shows FTIR spectra obtained for the VTMS coating on the titanium Grade 2 (a) and titanium alloy Ti13Nb13Zr (b) substrates. For both substrate materials, peaks occur at identical wavenumber values; the only difference shows up in their absorbance. The spectra of the VTMS coating show characteristic C-H stretching bands for the following wavenumbers: 3028, 2951 and 2844 cm^−1^ [30]. Bands recorded at the wavenumbers 1597 and 1408 cm^−1^ relate to the non-conjugate C=C bond. The peak revealed for the wavenumber 1275 cm^−1^ occurs due to the bending of the C-H bond [31]. The bands occurring in the range 975–1200 cm^−1^ are most likely associated with the stretching of the Si-O bond in the Si-O-Si or Si-O-C group [30,32]. As for the peak recorded at 961 cm^−1^, it indicates the stretching of either Si-O or Si-O-C in the Si-O-CH_3_ group [31]. The band occurring for the wavenumber 888 cm^−1^ is likely to originate from the Si-O-Ti bond [33]. The spectrum considered in [30] suggests that the peak recorded for the wavenumber 748 cm^−1^ may be associated with the C-H bond. In turn, the bands for wavenumbers from 400 to 700 cm^−1^ most probably originate from the stretching vibration band of Ti-O-Ti [33].

### 3.5. Adhesion of the VTMS Coating to the Substrate 

The use of Scotch^TM^ adhesive tape enabled an assessment of the adhesion of the VTMS coating to the substrate. Both on the Ti Gr2 substrate and on the Ti13Nb13Zr substrate, the coating exhibited very good adhesion to the substrate.

### 3.6. Corrosion Resistance Tests in a Bromide Ion Medium

For the VTMS coatings deposited on the Ti Gr 2 and Ti13Nb13Zr substrates, and for the same samples with no coating (model samples), potentiodynamic polarization curves were recorded in the potential range from −1.5 to +3.0 V in a 1 mol dm^−3^ NaBr solution. Figure 6 shows curves recorded for Ti Gr 2, while Figure 7 shows curves obtained for Ti13Nb13Zr. The recorded potentiodynamic curves confirm that the VTMS coating provides anodic and barrier protection for both the Ti Gr 2 and Ti13Nb13Zr substrates. The corrosion potential of the coating for both titanium Grade 2 and titanium alloy Ti13Nb13Zr shifts by approximately 0.53 V towards positive values. The recorded graphs show that titanium alloy Ti13Nb13Zr is characterized by greater corrosion resistance, as the corrosion potential of the uncoated titanium alloy Ti13Nb13Zr is −0.64 V, whereas that of Grade 2 is −1.13 V. When analyzing the samples covered with the VTMS coating, it can be noticed that the corrosion potential of titanium alloy Ti13Nb13Zr is −0.11 V, while that of Grade 2 is −0.60 V. Moreover, for both substrate materials, a reduction in cathodic current densities by two orders of magnitude and anodic current densities by three orders of magnitude occurred. 

The analyses of the structure and the potentiodynamic polarization curves confirm that both titanium Grade 2 and titanium alloy Ti13Nb13Zr not covered with a coating underwent pitting corrosion. The applied VTMS coating provided protection for titanium Grade 2 and titanium alloy Ti13Nb13Zr against pitting corrosion. To verify the resistance of the coating to pitting corrosion, the chronoamperometric method was employed. This method consists in recording variations in current density as a function of time after applying a potential to the working electrode. The pitting initiation potential, as seen in Figure 6, was +1.7 V. The measurements were performed in a 1 mol dm^−3^ NaBr solution. The recorded chronoamperometric curves are shown in Figure 8. It follows from the analysis of the chronoamperometric curves that the produced coating provided excellent corrosion protection to the substrate, as the current density stayed at a constant level for about 12 days. No pitting corrosion was observed during that time.

### 3.7. Corrosion Resistance Tests in a Chloride Ion Medium

For each coating, potentiodynamic curves were recorded in the potential range from −1.5 to +3.0 V in corrosion media simulating physiological fluids. Each of the solutions had a different chloride ion concentration. The tests were conducted in several solutions to consider the behavior of the coatings in the presence of different ions occurring in the human body. Thanks to this, a broad view of the influence of cations and anions on the produced coating was acquired. The composition of Ringer’s solution included the cations of sodium, potassium and calcium, and chloride anions. The composition of Hank’s fluid was extended by the presence of magnesium cations and hydrogencarbonate [HCO_3_^−^], dihydrogenphosphate (V) [H_2_PO_4_^−^], hydrogenphosphate (V) [HPO_4_^2−^] and sulphate (VI) [SO_4_^2−^] anions. As compared with Ringer’s solution, the composition of artificial saliva was extended by the dihydrogenphosphate (V) anion and the sulphate anion, as well as by components such as potassium rhodanate and urea. The solution that is most commonly used in electrochemical tests and, at the same time, contains the highest concentration of chloride ions and the broadest spectrum of components is simulated body fluid (SBF). In addition to the ions contained in Ringer’s solution, it contains magnesium cations and hydrogencarbonate, hydrogenphosphate (V) and sulphate (VI) anions, but it is also distinguished by the presence of tris(hydroxymethyl)aminomethane and hydrogen chloride. 

Table 2 shows the corrosion solutions used while considering the concentrations of chloride ions that they contain.

Figure 9 illustrates the corrosion behaviors of the VTMS coating on the titanium Grade 2 substrate, whereas Figure 10 depicts the corrosion behavior of the same coating on the titanium alloy Ti13Nb13Zr substrate, as dependent on the corrosion solution used. Despite the presence of chloride ions in the corrosion solutions, no pitting corrosion was found. The best shift in corrosion potential was obtained for the coating measured in the artificial saliva solution; however, the artificial saliva contained the lowest chloride ion concentration, as compared to the other solutions. The smallest shift in corrosion potential was obtained for the coating deposited on titanium Grade 2 and titanium alloy Ti13Nb13Zr, measured in Hank’s solution. In view of the above, the values of the corrosion potential shift were determined for the coating in the artificial saliva solution as compared to Hank’s solution. On the titanium Grade 2 substrate, a shift in corrosion potential by +1.23 V occurred, whereas on the titanium alloy Ti13Nb13Zr substrate, the corrosion potential was shifted by +1.09 V. The greatest reduction in current densities, both anodic and cathodic, was obtained for a sample with the coating deposited on titanium Grade 2 analyzed in Ringer’s solution, while the smallest was obtained for a sample in Hank’s solution. In the case of titanium alloy Ti13Nb13Zr, the greatest reduction in cathodic current densities was obtained for a sample analyzed in Ringer’s solution, while that of anodic current densities was obtained for a sample examined in the artificial saliva solution. Considering it contained the highest concentration of chloride ions, the best results were obtained for the simulated body fluid. Figure 11 depicts the structure of the titanium Grade 2 substrate (after the removal of the coating) upon completion of corrosion tests in the simulated physiological fluid. The photographs confirm that no pitting corrosion of the metal substrate occurred.

## 4. Conclusions

Coatings based on vinyltrimethoxysilane were produced on the substrates of titanium Grade 2 and titanium alloy Ti13Nb13Zr using the sol-gel method. The dip coating method was employed for their application on the substrate surface. The concentration of the main component of the coatings—vinyltrimethoxysilane—was 3.92 mol dm^−3^.

The investigation carried out and the analysis conducted allow us to draw the following practical conclusions:The application of vinyltrimethoxysilane on titanium Grade 2 and titanium alloy Ti13Nb13Zr substrates resulted in the formation of highly adhesive coatings which protected the material against corrosion in various media.The obtained coatings did not show any cracks and discontinuities and were homogeneous.The coating produced by the sol–gel method uniformly covered the substrate surface and did not show large differences in thickness. The thickness of the obtained coating ranged from 9.5 to 16.7 μm.The performed examination showed a low degree of surface roughness of the obtained coating, which makes it extremely attractive from the point of view of corrosion resistance. Too high a surface roughness of coatings favors the development of pitting corrosion on the surface of biomaterials, especially in hollows or depressions where the coating is the thinnest.FTIR spectroscopy revealed the following bonds to be present in the coating: C-H, C=C, Si-O, Si-O-C, Si-O-Ti and Ti-O-Ti.Corrosion resistance tests were carried out in a solution containing bromide ions, as well as in simulated physiological solutions in the presence of chloride ions. As shown by the investigation, the VTMS coatings applied to the substrates of titanium Grade 2 and titanium alloy Ti13Nb13Zr offered corrosion protection.The vinyltrimethoxysilane-based coatings stabilized the corrosion potential within the passive state (anodic protection) and provided barrier protection.Corrosion tests carried out in a sodium bromide solution showed no pitting corrosion.The recorded chronoamperometric curves confirmed the resistance of the coating to pitting corrosion. Tests carried out in simulated physiological solutions demonstrated that the vinyltrimethoxysilane-based coating produced by the sol–gel method, as proposed in this paper, significantly enhanced the corrosion resistance of the investigated materials, which confirms its effectiveness and potential for being applied in medicine, for example, in implantology.Based on the obtained results of tests for, among others, surface roughness and corrosion resistance, it can be stated that VTMS coatings can be used for covering knee or hip implants.

## Figures and Tables

**Figure 1 materials-14-06350-f001:**
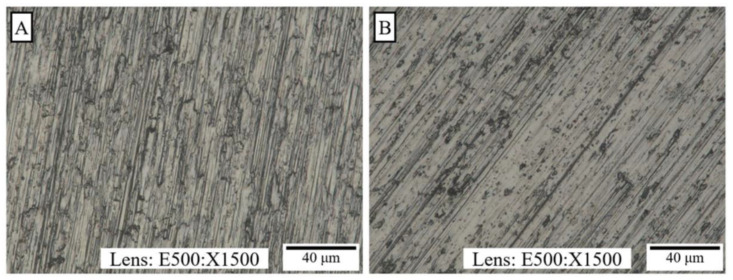
Structure of the VTMS coating on the Ti Gr 2 (**A**) and Ti13Nb13Zr (**B**) substrates. Photos were taken with a KEYENCE VHX digital microscope.

**Figure 2 materials-14-06350-f002:**
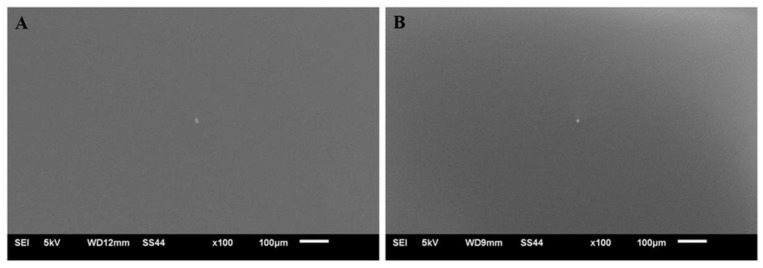
VTMS coating on the Ti Gr 2 (**A**) and Ti13Nb13Zr (**B**) substrates. Photos were taken with a JEOL JSM-6610 LV scanning electron microscope.

**Figure 3 materials-14-06350-f003:**
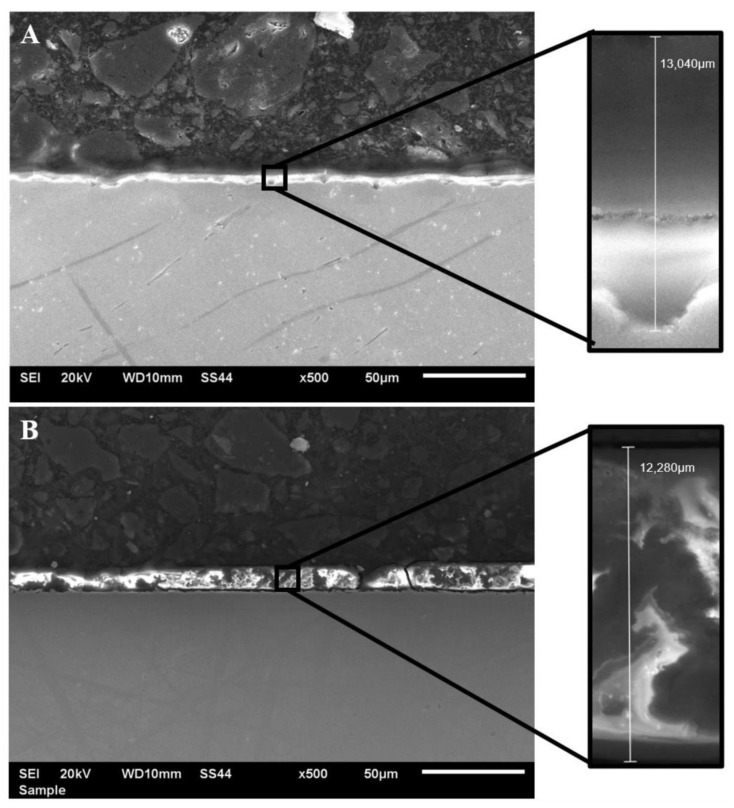
Cross-section of the VTMS coating deposited on titanium Grade 2 (**A**) and Ti13Nb13Zr titanium alloy (**B**) captured with a JEOL JSM- 6610 LV scanning electron microscope.

**Figure 4 materials-14-06350-f004:**
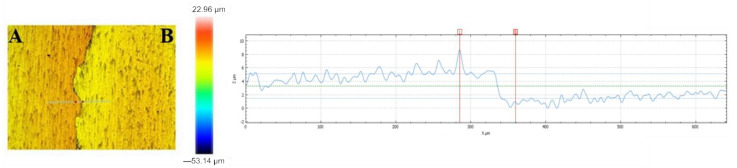
Profile of the VTMS coating (**A**) deposited on titanium alloy Ti13Nb13Zr in relation to the substrate without the coating (**B**), recorded with a SENSOFAR profilometer.

**Figure 5 materials-14-06350-f005:**
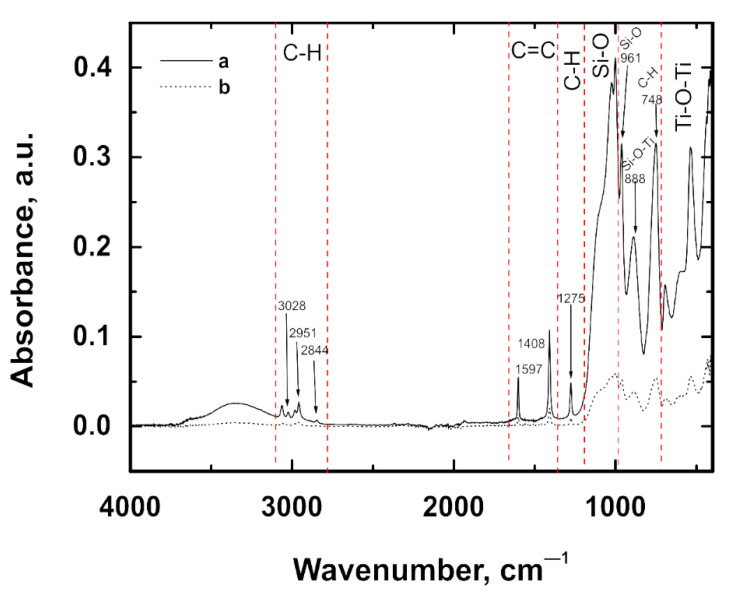
FTIR spectra obtained for VTMS coatings deposited on the Ti Gr 2 (a) and Ti13Nb13Zr (b) substrates.

**Figure 6 materials-14-06350-f006:**
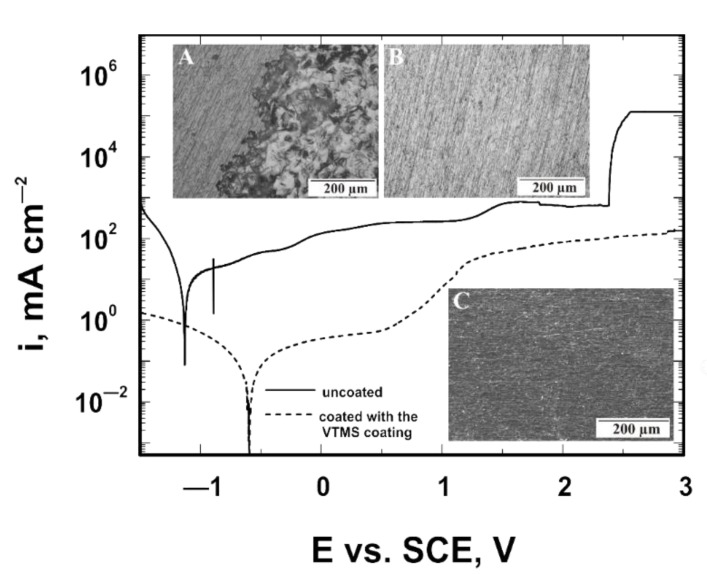
Potentiodynamic polarization curves analyzed in 1 mol dm^−3^ NaBr solution for Ti Gr 2 with the microstructure of the uncoated substrate (**A**) and substrate after removing the VTMS coating (**B**,**C**) after corrosion tests. A and B were taken with the GX41 Olympus optical microscope, whereas C was taken with a JEOL JSM- 6610 LV scanning electron microscope.

**Figure 7 materials-14-06350-f007:**
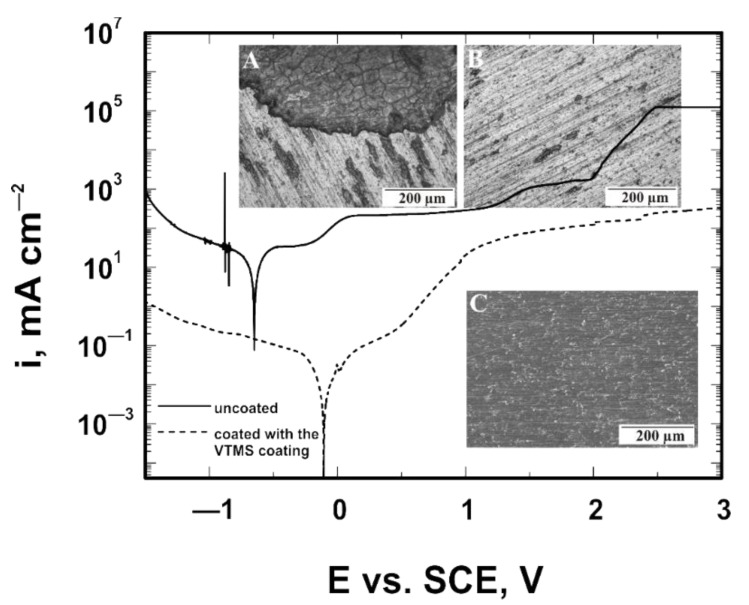
Potentiodynamic polarization curves analyzed in 1 mol dm^−3^ NaBr solution for Ti13Nb13Zr with the microstructure of the uncoated substrate (**A**) and substrate after removing the VTMS coating (**B**,**C**) after corrosion tests. A and B were taken with the GX41 Olympus optical microscope, whereas C was taken with a JEOL JSM-6610 LV scanning electron microscope.

**Figure 8 materials-14-06350-f008:**
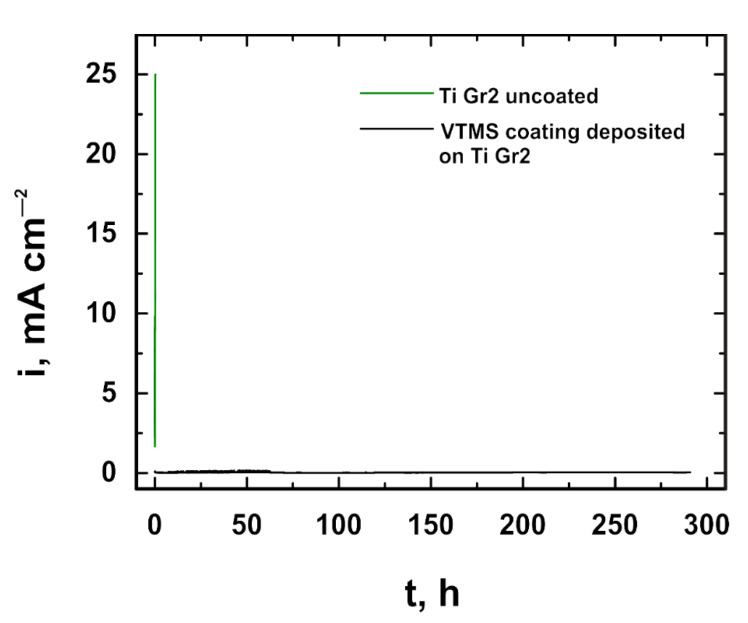
Chronoamperometric curves recorded in a 1 mol dm^−3^ NaBr solution at a potential of +1.7 V.

**Figure 9 materials-14-06350-f009:**
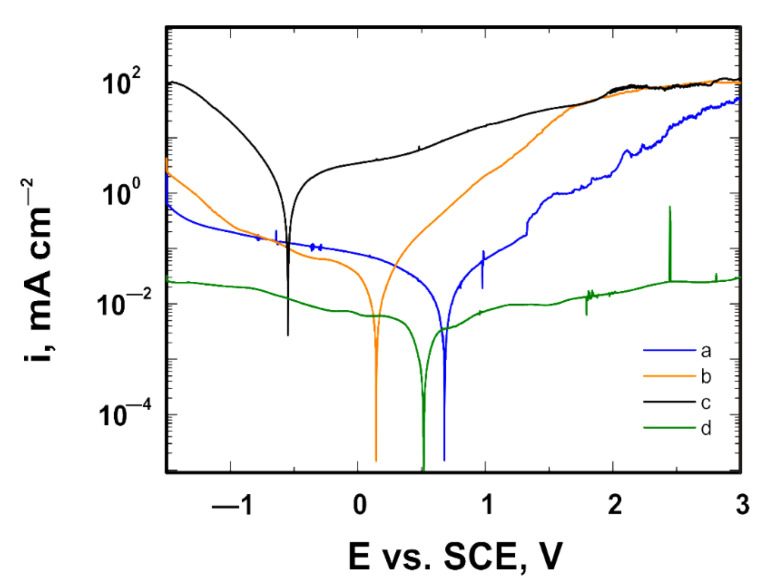
Potentiodynamic polarization curves analyzed in (a) artificial saliva solution, (b) simulated body fluid, (c) Hank’s solution and (d) Ringer’s solution, for Ti Gr 2 coated with the VTMS coating.

**Figure 10 materials-14-06350-f010:**
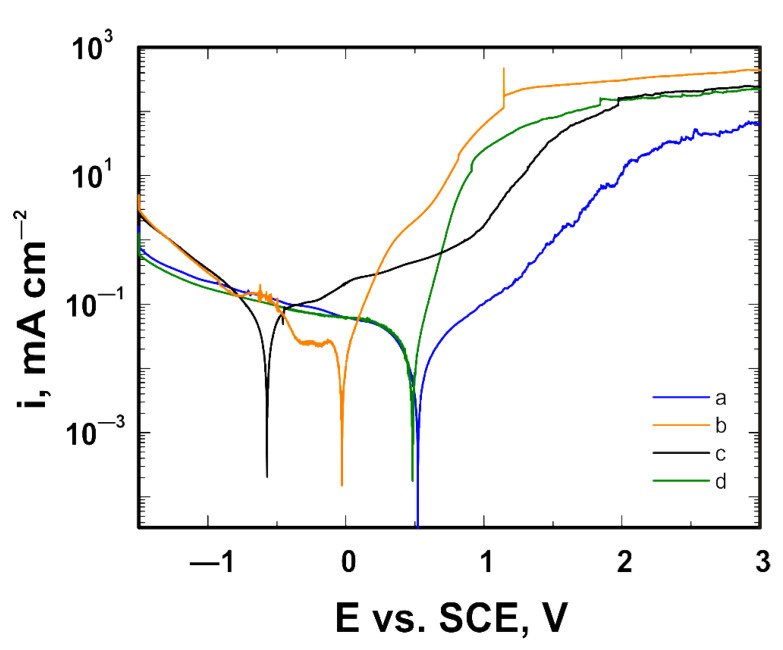
Potentiodynamic polarization curves analyzed in (a) artificial saliva solution, (b) simulated body fluid, (c) Hank’s solution and (d) Ringer’s solution, for Ti13Nb13Zr coated with the VTMS coating.

**Figure 11 materials-14-06350-f011:**
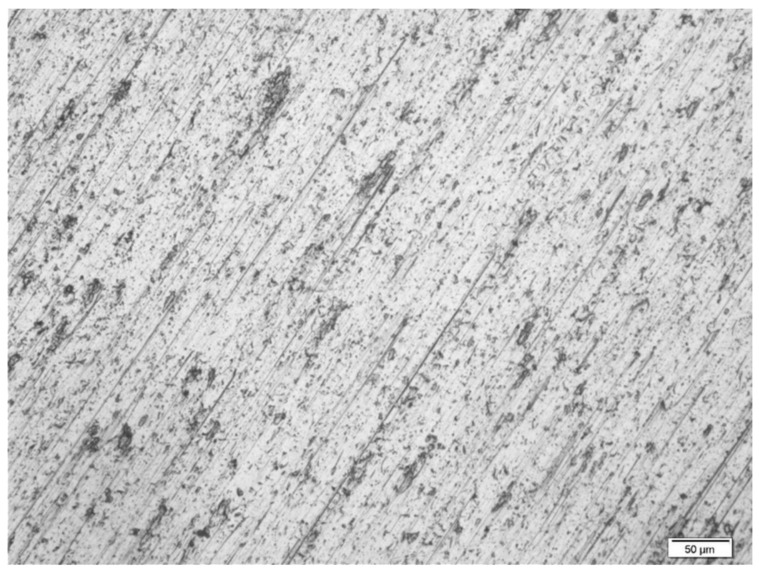
Structure of Ti Gr 2 (after removal of the coating) after corrosion tests in simulated body fluid. The photo was taken with the GX41 Olympus optical microscope.

**Table 1 materials-14-06350-t001:** Chemical composition of physiological fluids.

	Ringer’s Fluid [10]	Hank’s Fluid [27]	Simulated Body Fluid [28]	Artificial Saliva Solution [29]
NaCl	8.6 g dm^−3^	8 g dm^−3^	8.035 g dm^−3^	0.4 g dm^−3^
KCl	0.3 g dm^−3^	0.4 g dm^−3^	0.225 g dm^−3^	0.4 g dm^−3^
CaCl_2_	0.243 g dm^−3^	0.14 g dm^−3^	0.292 g dm^−3^	0.6 g dm^−3^
NaHCO_3_	–	0.35 g dm^−3^	0.355 g dm^−3^	–
KH_2_PO_4_	–	0.06 g dm^−3^	–	–
MgCl_2_·6H_2_O	–	0.1 g dm^−3^	0.311 g dm^−3^	–
Na_2_HPO_4_·2H_2_O	–	0.06 g dm^−3^	–	–
MgSO_4_·7H_2_O	–	0.06 g dm^−3^	–	–
K_2_HPO_4_·3H_2_O	–	–	0.231 g dm^−3^	–
Na_2_SO_4_	–	–	0.072 g dm^−3^	–
((HOCH_2_)_3_CNH_2_)	–	–	6.118 g dm^−3^	–
HCl (1 mol dm^−3^)	–	–	39 ml dm^−3^	–
NaH_2_PO_4_·2H_2_O	–	–	–	0.26 g dm^−3^
KSCN	–	–	–	0.3 g dm^−3^
Na_2_S·9H_2_O	–	–	–	0.005 g dm^−3^
urea	–	–	–	1 g dm^−3^

**Table 2 materials-14-06350-t002:** Chloride ion concentration in individual simulated body fluids.

Corrosive solution	Ringer’s Solution	Hank’s Solution	Simulated Body Fluid	Artificial Saliva Solution
Concentration of chloride ions	0.16 mol dm^−3^	0.15 mol dm^−3^	0.19 mol dm^−3^	0.02 mol dm^−3^

## Data Availability

The data presented in this study are available on request from the corresponding author.

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
