# Peer review of "The Effect of the Corrosion Medium on Silane Coatings Deposited on Titanium Grade 2 and Titanium Alloy Ti13Nb13Zr"

_materials, 2021, doi:10.3390/ma14216350_

Round 1
Reviewer 1 Report
- In the abstract section, the authors expressed, they investigated the adhesion of the coating to the substrates. However, in the main body and conclusion of the manuscript, there is no explanation for this.
- In the introduction section, the authors should mention the other applications of titanium alloys. For instance, they should explain the demand for using titanium alloys as a substitute for aluminum alloys in aerospace applications is growing. This is because some researchers have shown that even a normal humidity of the environment at room temperature can lead to a significant ductility loss in aluminum alloys [https://doi.org/10.1016/j.vacuum.2021.110489].
- The font size of the written text in Figure 1 is so small.
- Further explanation is required for Figure 2.
- For a better comparison of the coating thicknesses, it is recommended to add some SEM images showing the coating cross-section in the Ti13Nb13Zr alloy.
- The font size of the written texts in Figures 6 and 7 is also so small.
- The resolution of Figure 11 needs to be improved. Also, please change Figure 10 to Figure 11 in the caption of Figure 11.
- It is recommended to use SEM micrographs of both alloys to show the surfaces after removal of the coatings.
- The corrosion behaviors of the titanium Grade 2 and Ti13Nb13Zr coated with VTMS in the same solution are different (Figures 9 and 10). Why?
Author Response
Dear Reviewer,
We are grateful for your remarks regarding our manuscript.
In this letter we would like to address the issues you pointed out in your review.

Reviewer 2 Report
Dear Authors,
I have read your paper "The effect of the corrosion medium on silane coatings deposited on titanium Grade 2 and titanium alloy Ti13Nb13Zr" carefully.
This paper describes the effect of the corrosion medium on silane coatings deposited on titanium aloys
The paper is easy to read.
But the methods are not properly described, so that other research groups may not reproduce them.
The paper is interesting. However, it requires few corrections.
1. Please describe the chemical composition of the titanium Grade 2 and titanium alloy Ti13Nb13Zr
2. There is no information about adhesion to the substrate
3. Also, there is no information about method determination of the adhesion to the substrate
4. The discussion should be extended. Now the discussion is poor
The paper can be accepted for publication only after major improvements.
Author Response

(The authors gave the same response as above.)

Reviewer 3 Report
The work entitled ‘The effect of the corrosion medium on silane coatings deposited on titanium
Grade 2 and titanium alloy Ti13Nb13Zr’ describes how a chemical method (silanization) has been
applied to create a corrosion resistant coating on Ti grade 2 and Ti13Nb13Zr. The electrochemical investigation (potentiodynamic polarization) of the coated substrates is performed after their immersion in several artificial body fluids. Surface structure results of substrates obtained with SEM microscopy are also presented, together with the FTIR analyses.
- The length bars in Figure 1, micrographs (A) and (B) should be improved; even with magnification it is difficult to read them.
- Check the English and the spelling errors: e.g., ‘Figure 2 shows photographs confirming that the applied coating overs uniformly the...’
- “The photographs shown in Figure 6 and Figure 7 were taken with a GX41 Olympus optical microscope.” should be in the Section of Materials and Methods.
- A phrase should be added in the Section of Conclusions, to mention specific applications in biomedicine (e.g., hip), based on the corrosion resistance results.
Author Response

(The authors gave the same response as above.)

Round 2
Reviewer 1 Report
Comparison of titanium with other alloys, especially aluminum alloys, shows the superiority of titanium in many aspects. It would have been better if this comment had been considered.
Reviewer 2 Report
Dear Authors,
I have read your modified paper "The effect of the corrosion medium on silane coatings deposited on titanium Grade 2 and titanium alloy Ti13Nb13Zr" carefully.
The materials and methods are properly described, so that other research groups may reproduce them. Explanations are clear and the paper is easy to read.
I can recommend the Editor to accept this revised manuscript to be published in Materials